# An inorganic-rich but LiF-free interphase for fast charging and long cycle life lithium metal batteries

Muhammad Mominur Rahman[1], Sha Tan [1], Yang Yang [2], Hui Zhong[3], Sanjit Ghose[2], Iradwikanari Waluyo [2], Adrian Hunt[2], Lu Ma[2], Xiao-Qing Yang [1] & Enyuan Hu [1] ✉

Li metal batteries using Li metal as negative electrode and $LiNi_{1-x-y}Mn_xCo_yO_2$ as positive electrode represent the next generation high-energy batteries. A major challenge facing these batteries is finding electrolytes capable of forming good interphases. Conventionally, electrolyte is fluorinated to generate anion-derived LiF-rich interphases. However, their low ionic conductivities forbid fast-charging. Here, we use $CsNO_3$ as a dual-functional additive to form stable interphases on both electrodes. Such strategy allows the use of 1,2-dimethoxyethane as the single solvent, promising superior ion transport and fast charging. $LiNi_{1-x-y}Mn_xCo_yO_2$ is protected by the nitrate-derived species. On the Li metal side, large $Cs^+$ has weak interactions with the solvent, leading to presence of anions in the solvation sheath and an anion-derived interphase. The interphase is surprisingly dominated by cesium bis(fluorosulfonyl)imide, a component not reported before. Its presence suggests that $Cs^+$ is doing more than just electrostatic shielding as commonly believed. The interphase is free of LiF but still promises high performance as cells with high $LiNi_{0.8}Mn_{0.1}Co_{0.1}O_2$ loading (21 mg/cm$^2$) and low N/P ratio (~2) can be cycled at 2C (~8 mA/cm$^2$) with above 80% capacity retention after 200 cycles. These results suggest the role of LiF and Cs-containing additives need to be revisited.

Li metal batteries (LMB) hold great promise for high energy applications such as electric vehicles[1]. Li metal has very high theoretical capacity (3860 mAh/g) and the lowest redox potential (−3.04 V vs Standard hydrogen electrode)[2,3]. When coupled with high Ni content $LiNi_{0.8}Mn_{0.1}Co_{0.1}O_2$ (NMC811)[4], the system offers significantly high energy density. The electrolyte being responsible for transporting ions between the electrodes, is a key component in lithium metal batteries as it must have the capability to form stable interphases on both electrodes, which has been found to be very challenging[5]. Various electrolyte design strategies have been explored including high concentration electrolytes[6,7], localized high concentratison electrolytes[8–11],

and solvent molecule tuning[5,12]. These novel electrolytes have enabled LMB with long cycle even under harsh conditions such as high NMC loading and low N/P ratio. The N/P ratio here refers to the capacity ratio between the negative electrode (Li) and the positive electrode (NMC) in batteries.

While significant progress has been made in LMB electrolyte development, LMB still faces the grand challenge of fast charging. In the state-of-the-art LMB electrolytes mentioned previously, fluorinated solvents are either the dominating or the only solvent components[5,9]. On the one hand, these fluorinated solvents have weak solvation interaction with the lithium cation, leading to fast

[1]Chemistry division, Brookhaven National Laboratory, Upton, NY 11973, USA. [2]National Synchrotron Lightsource II, Brookhaven National Laboratory, Upton, NY 11973, USA. [3]Department of Joint Photon Sciences Institute, Stony Brook University, Stony Brook, NY 11970, USA. ✉e-mail: enhu@bnl.gov

desolvation process and an interphase that is formed by the desired anion decomposition[13,14]. On the other hand, the exact weak solvation compromises the solubility of lithium salt and hence impedes the ion transport in the bulk electrolyte[15–17]. The result is that LMB has to be charged at a slow rate for stable cycling.

An electrolyte that provides fast charging capability for LMB has to have good ion transport property[18]. The interphase stability issue may be addressed by an electrolyte additive rather than the electrolyte solvent[19–21]. This separates the interphase formation functionality from the ion transport one and the solvents do not have to be fluorinated at the risk of being incapable of transporting lithium ions in the bulk electrolyte. In designing the appropriate electrolyte additive, it is important to bear the crosstalk effect in mind[22–24]. It has been becoming increasingly clear that one electrode influences the interphase formation process in the other. For example, our recent study shows NMC can also catalyze the decomposition of salt and form anion-derived species on the lithium metal side[25]. Research by others indicates negative electrode chemistry (graphite, lithium, or lithium titanate) can also influence the positive electrode interphase formation[26–28]. These results suggest that the electrodes cannot be treated separately. Instead, they should be treated as a system, and a good protection is only possible when both the positive and negative electrodes are stabilized simultaneously. This emphasizes the dual protection strategy. Several studies have explored the strategy with various electrolyte additives[29,30]. We will further explore the strategy through rational design of single additive that can simultaneously act upon both electrodes to promote the formation of stable interphases.

Cs-containing salts have been used before as LMB electrolyte additives and it has been believed that Cs+ does not contribute to interphase formation but serve as electrostatic shield on lithium metal which promotes lithium plating uniformity. Unfortunately, none of the previously proposed Cs-containing salts can really enable LMB under pratical conditions. We believe the reasons are 1. there is lack of consideration of dual-protection strategy; 2. the working mechanism of Cs+ needs to be revisted. In this work, we address the first issue by using CsNO3 which we expect to protect the electrodes simultaneously. The second issue is addressed by recent advancement in interphase characterization which can provide accruate crystalline and amorphous information about the lithium metal interphase. We show that Cs+ leads to an SEI dominated by the inorganic species cesium bis(fluorosulfonyl)imide (CsFSI), a component that has never been reported. Interestingly, this SEI is free of LiF which has long been regarded as a necessary component for good interphase. The quality of the CsFSI-dominant, LiF-free interphase is supported by the superior electrochemical performance both in terms of long cyclce life and fast-charging capabilites.

## Results

### Dual functionality of CsNO₃ additive

Low concentration nonfluorinated ether electrolyte has the advantage of low cost but suffers from instability issues on the electrodes[31]. As mentioned in the introduction section, it is important to protect both the electrodes simultaneously considering the crosstalk effect. Cs-containing additives have been reported before with interesting interphase protection functionalities[32,33]. A possible electrolyte additive candidate that serves such a purpose is CsNO3. Under the influence of electric field, Cs+ and NO3- will be attracted to the electron-rich Li metal surface (negative electrode) and the electron-deficient NMC811 surface (positive electrode), respectively, initiating the formation of protective interphases on both electrodes. An illustration of such a design principle is shown in Fig. 1a. Figure 1b–e indicates CsNO3 worked in the way it is designed to function. Figure 1b is the Cs L3-edge fluorescence X-ray absorption spectroscopy for NMC811 and Li metal from the cycled cell. As the fluorescence signal is proportional to the Cs concentration, the fact that the peak intensity is much more

prominent for the Li metal than that for the NMC811 indicates that Cs element is mostly found in the Li metal interphase. The peak energy (around 5015.9 eV) suggests that Cs is in the Cs+ form in the interphase, not as Cs metal[34]. The enrichment of Cs on the Li metal side is quantitatively understood by X-ray fluorescence (XRF) imaging mapping of the whole electrode (Fig. 1c). The Cs concentration in the Li metal interphase is around 92 μg/cm² while that in the NMC811 interphase is only around 4 μg/cm² after the same number of cycles. The accumulation of Cs is gradual, but majority of the Cs accumulates on the Li metal by the first 50 cycles (Supplementary Fig. S1). These results confirm that a concentration gradient of Cs+ is developed in the electrolyte and Cs+ mostly goes to the Li metal side. Figure 1d, e suggests a concentration gradient is also developed for the NO3- and they mostly go to the NMC811 side. Nitrogen K-edge spectra in Fig. 1d show the presence of NO3- derived species in the NMC811 interphase[35,36]. Similar components have also been observed for the interphase resulting from electrolyte containing LiNO3 additive (Supplementary Fig. S2). N based species is also observed on the Li metal side, but the spectrum shows distinct features from that for the interphase on the NMC811 side (Fig. 1e). Comparison with the N spectrum of LiFSI salt suggests the possible presence of FSI anions in the Li metal interphase. Detailed characterization and discussion of the Li metal interphase components will be presented later. In summary, the dual protection design strategy illustrated in Fig. 1a has been well executed. When using CsNO3 as an additive for the low concentration LiFSI in DME electrolyte, Cs+ tends to accumulate on the Li metal side whereas NO3- accumulates on the NMC811 side. Cs+ and NO3- are expected to play their protective roles on the Li metal and the NMC811, respectively. The detailed protection mechanism is discussed as follows.

### An SEI dominated by CsFSI and absent of LiF

The interphase usually consists of organic and inorganic components, both of which are formed by electrolyte decomposition[37]. In recent years, our group has developed synchrotron X-ray diffraction (XRD) and pair distribution function (PDF) analysis to characterize both the crystalline and amorphous components of the interphase[37]. A series of new findings have been obtained[25] and several of them have already been supported and confirmed by independent studies by other research groups[38–40]. The XRD pattern of the solid-electrolyte interphase (SEI) of Li metal resulting from the baseline electrolyte is shown in the upper panel of Fig. 2a. Thanks to the high quality of synchrotron XRD, Rietveld refinement can be carried out reliably. It has typical SEI components such as LiH, Li2O, and nanocrystalline LiF that result from electrolyte decomposition (Supplementary Table S1). It should be noted that LiH and LiF have the same crystal structure but different lattice parameters[41–43]. The difference is significant enough to be determined by Rietveld refinement. When LiH and LiF are both present in the SEI, they usually have different microstructure properties, with LiH being in the form of large crystals (>200 nm) and LiF being nanocrystals (~10 nm)[40,44]. More details on LiH identification in the interphase can be found in our previous work[37]. In sharp contrast, the SEI resulting from the electrolyte with CsNO3 additive is completely different as shown in the lower panel in Fig. 2a. Instead of Li2O and LiF, CsFSI is the dominant inorganic species in the SEI (Supplementary Table S2). Detailed Rietveld refinement indicates that there are two kinds of CsFSI present, one being cis-CsFSI (major) and the other being trans-CsFSI (minor) (lattice models in Fig. 2). These two structures have previously been studied by Matsumoto and Farrugia from a crystallographic point of view[45,46]. They have similar crystal packing modes but differ in the arrangement of atoms in the FSI anion. The formation of CsFSI may not consume electrons from the circuit compared with the formation of Li2O and LiF, both of which do consume electrons. As a result, CsFSI formation may be favorable for improving Coulombic efficiency. Interestingly, there is a dynamic evolution of the two isomorphs of CsFSI upon cycling. Comparison between the SEI species

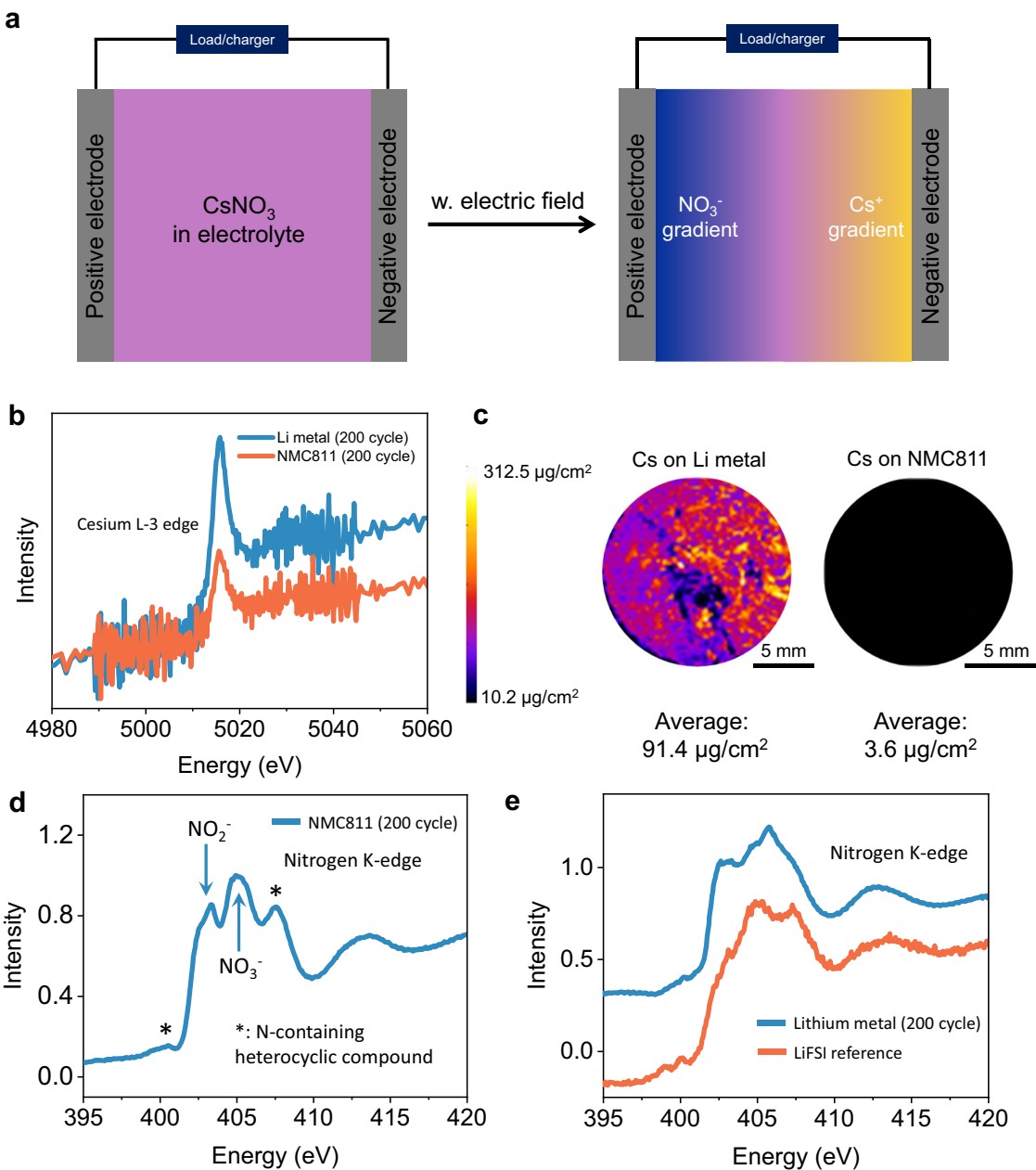

**Fig. 1 | Working mechanism of the dual functionality of CsNO₃ additive.**
**a** Schematic illustration of the concentration gradient development of the Cs⁺ and NO₃⁻ species when voltage is applied in a Li metal cell. **b** Cs L-3 edge spectra of NMC811 and Li metal after cycling in the electrolyte containing 3 wt% CsNO₃ additive. **c** X-ray fluorescence (XRF) images showing the distribution of Cs species on the surface of NMC811 and Li metal after cycling in electrolyte (200 cycles) containing 3 wt% CsNO₃ additives. **d, e** Show the N K-edge spectra of NMC811 and Li metal, respectively after cycling in electrolyte containing 3 wt% CsNO₃ additive. The cycling was performed at C/2 rate (0.8 mA/cm²) in high N/P cells.

after 100 cycles (Supplementary Fig. S3) and 200 cycles (bottom panel of Fig. 2a) shows that upon cycling, the fraction of cis-CsFSI increases while that of trans-CsFSI decreases, indicating a trans-to-cis phase transition and the cis- isomorph being the more thermodynamically stable one. This is consistent with previous findings reported in the literature[46]. While XRD data reveals the crystalline components in the SEI, PDF data provides information on the amorphous phases (Fig. 2b). For the baseline electrolyte, the first PDF peak is attributed to the presence of S=O, S-F, or S-N bonds (around 1.5 Å) that belong to species resulting from FSI⁻ anion decomposition. The calculated structure and PDF pattern of a model compound $Li_2(FSI_{(-F)})_2$ are shown for illustration. The first PDF peak is asymmetric and has some intensity in the lower $r$ region, suggesting the presence of some shorter bonds which are likely carbon-based bonds such as C-O, C=O, and C-C. These

bonds belong to species such as DME oligomer that result from solvent decomposition. The calculated structure and PDF data of DME oligomer are also shown in the figure. Other PDF peaks can be well accounted for by the presence of interphase components such as lithium metal, LiH, $Li_2O$, and LiF. The peak at 2.0 Å can only be from inorganic species such as LiH, $Li_2O$, and LiF. Its relative strong intensity suggests that there are abundant inorganic species in the SEI formed by the baseline electrolyte decomposition. Following the same analysis method, the lower panel of Fig. 2b suggests that the amorphous phase resulting from FSI⁻ anion decomposition has a dominant percentage in the SEI formed by electrolyte containing CsNO₃ additive. The inorganic species in the SEI is CsFSI, with no $Li_2O$ or LiF present. The significant difference in SEI components may be caused by the solvation structure, which is influenced by the presence of Cs⁺ cation. As

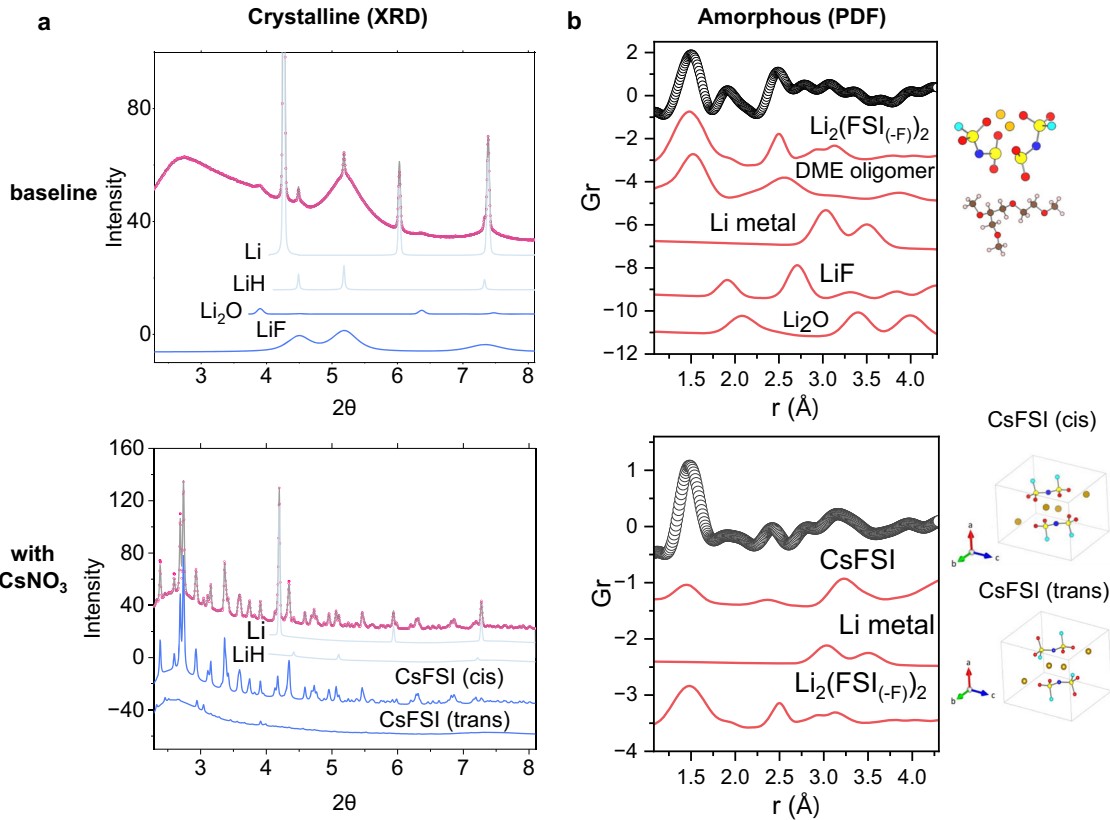

**Fig. 2 | Crystalline and amorphous components in the lithium metal interphase. a** Crystalline components of the solid-electrolyte interphase (SEI) of Li metal revealed by synchrotron XRD and Rietveld refinement. The red is the fit and the blue is the XRD pattern for individual phases calculated from Rietveld refinement. **b** Amorphous components of the SEI revealed by PDF studies. The scattered plot is the measured PDF data, and the red is the PDF pattern calculated from the possible individual components of the SEI[37]. SEI species were characterized after 50 cycles in the baseline electrolyte and after 200 cycles in the electrolyte containing the $CsNO_3$ additive. The cycling was performed at C/2 rate ($0.8\ mA/cm^2$) in high N/P cells. The color code for the atoms is the same in all subfigures: red: oxygen; blue: nitrogen; cyan: fluorine; brown: carbon; white: hydrogen; gold: lithium; dark yellow: cesium, light yellow: sulfur.

Supplementary Fig. S4 illustrates, in the baseline electrolyte, due to the relatively small size of the $Li^+$ cation and the strong solvation interaction between DME solvent and the $Li^+$ cation, $FSI^-$ anions have small chance of entering the solvation sheath and participating in the interphase formation. Size of the additive cation is also an important factor as the SEI of Li metal cycled with other alkali metal nitrate additives only shows regular SEI components (Supplementary Fig. S5). When the $Cs^+$ cation is present, its large size ($Li^+$ radius: 0.59 Å; $Cs^+$: 1.67 Å) greatly weakens the solvation interaction between DME solvent and $Cs^+$ cation. Also because of its large size, $Cs^+$ can be stabilized in a larger solvation sheath than that for the case of $Li^+$. As a result, more anions can enter the solvation sheath and contribute to the interphase formation. Such formed interphase contains both $FSI^-$ anion derived amorphous components and CsFSI inorganic species, which considerably improved the lithium stripping plating efficiency as will be discussed below.

### Electrochemical characterization of LMBs using $CsNO_3$-containing electrolyte

The effect of the CsFSI-rich SEI and the N-rich NMC811 interphase on electrochemical performance is evident from the cycling of Li metal cells. Both the baseline electrolyte and the electrolyte with the $CsNO_3$ additive show comparable oxidative/reductive decomposition potential (Supplementary Fig. S6). However, the interphase compositions resulting from the decomposition of these two electrolytes are significantly different as discussed in Fig. 2. The unique interphase formed in the $CsNO_3$ containing electrolyte positively influences the Coulombic efficiency (CE). CE of Li plating and stripping (Fig. 3a) is

tested in Li||Cu cells following the method proposed by Zhang and coworkers[47] for the baseline electrolyte and electrolyte with $CsNO_3$ additive (cycling details are in the experimental section). The electrolyte with $CsNO_3$ additive has a higher CE (99.46%) than the baseline electrolyte (99.11%) as shown in Fig. 3a in the Li||Cu cell. This indicates an enhanced stability of the Li metal in the $CsNO_3$-containing electrolyte. Li metal stability is further evaluated in a Li||Li symmetric cell at $1\ mA/cm^2$ current density with $1\ mAh/cm^2$ plating/stripping capacity. Li||Li symmetric cell cycling also shows an improved Li plating/stripping stability in the presence of the $CsNO_3$ additive (Supplementary Fig. S7). Li metal cells are constructed with Li metal and NMC811 to evaluate baseline electrolyte and electrolyte with $CsNO_3$ additive. Using the baseline electrolyte, cells under mild conditions (250 μm thick Li, $9\ mg/cm^2$ NMC811) can maintain stable cycling for up to 70 cycles (Fig. 3b and Supplementary Fig. S8a). However, a rapid capacity drop is observed afterwards. In contrast, the electrolyte containing the $CsNO_3$ additive significantly improved the cyclability of the cell under mild conditions. Around 70% capacity retention is maintained even after 300 cycles (Fig. 3b and Supplementary Fig. S8b). Furthermore, the CE of the full cell can reach as high as 99.5% after 40 cycles (Fig. 3d). It should be noted that other alkali metal (Li, Na, K, and Rb) nitrate additives also tested in this study show some improvement of the cycling stability of Li metal batteries but not to the extent of $CsNO_3$ (Supplementary Figs. S9–S11). Moreover, LiFSI in DME based electrolytes offer high ionic conductivity. For example, 1.5 M LiFSI salt concentration delivers reasonably high ionic conductivity without using too high salt concentration (Supplementary Fig. S12). The high ionic conductivity is still maintained with the $CsNO_3$ additive

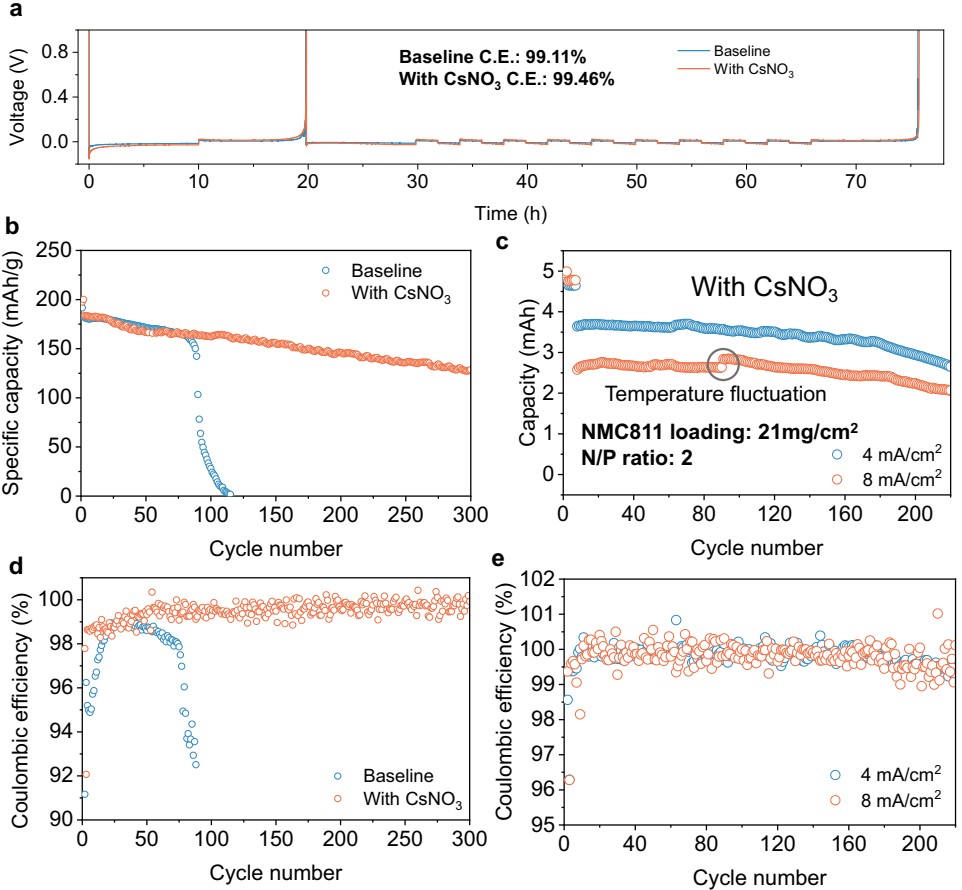

**Fig. 3 | Electrochemical cycling performance of Li metal batteries in baseline electrolyte and with CsNO3 additive. a** Coulombic efficiency measurement following the Aurbach method in a Li||Cu cell[47]. **b** Cycling stability of Li metal cell with NMC811 and Li metal with baseline electrolyte and electrolyte containing 3 wt% CsNO₃ additive. NMC811 loading is 9 mg/cm² and Li metal used is 250 µm thick. Cycling is performed at C/2 rate (0.8 mA/cm²). **c** Cycling stability of Li metal cell using NMC811 with 21 mg/cm² loading and 50 µm thick Li metal in the electrolyte containing 3 wt% CsNO₃ additive. The cells go through two formation cycles at a rate of C/10 (0.4 mA/cm²) and five cycles at C/5 rate (0.8 mA/cm²) before being cycled at 1 C (4 mA/cm²) rate and 2 C (8 mA/cm²) for later cycles. **d** Coulombic efficiency of the cells in Fig. 3b. **e** Coulombic efficiency of the cells in Fig. 3c.

(Supplementary Fig. S13). As mentioned in the introduction, charging current in fluorinated electrolytes is largely limited because of the low ionic conductivity, making them inoperable at high current density. However, since CsNO₃ additive can stabilize the electrode interphases, DME based electrolytes with CsNO₃ additives offer immense opportunity for stable high current cycling. Indeed, the dramatic effect of using CsNO₃ as the additive on fast charging can be clearly seen in Fig. 3c. LMB cells first go through two formation cycles at C/10 rate (0.4 mA/cm²) and 5 cycles at C/5 rate (0.8 mA/cm²) and then are cycled at 1 C rate (4 mA/cm²). Capacities from all these formation cycles are plotted in Fig. 3c. Even under harsh conditions (50 µm thick Li foil, 21 mg/cm² NMC811, N/P ratio = 2), cells using electrolytes with the CsNO₃ additive maintain more than 80% of the initial capacity even after 200 cycles (Fig. 3c and Supplementary Fig. S14a) while the cell using the baseline electrolyte quickly loses its capacity within 20 cycles (Supplementary Figs. S15 and S16). The CE of the full cell reaches around 99.7% after several initial cycles with the additive (Fig. 3e). In some cycles (both in Fig. 3d, e), the CE may even go up over 100%. Even though CE is routinely measured and conventionally regarded as the indicator of efficiency, it should be noted that the real meaning of CE is the number of electrons flow during discharge divided by the electrons flow during charge. Due to the presence of side reactions, there is no direct correlation between the number of electrons flow and the number of lithium ions transported. As a result, CE may not be a good indicator of efficiency for LMB and the randomness of side

reactions during charge-discharge may give rise to CE number over 100%. For example, more electrons could be consumed during discharge than during charge because of more side reactions and this would lead to a CE greater than 100%. Stable cycling is observed at an even higher charging rate of 8 mA/cm² in low N/P cells with the CsNO₃ additive, maintaining more than 80% capacity after 220 cycles (Fig. 3c and Supplementary Fig. S14b). A comparison between the cycling current density and cycle life of some of the state-of-the-art electrolytes reported in the literature indicates that CsNO₃ containing DME based electrolyte can combine both good cyclability with high current density (Supplementary Fig. S17 and Supplementary Table S4). Such stable cycling is obtained without following the popular fluorination strategy of electrolyte. These results indicate that both the electrodes must have been well protected during the cycling, validating the dual protection strategy proposed in the beginning of this paper.

## Dense and homogeneous lithium deposition enabled by the desired interphase

The scanning electron microscopy (SEM) images of the cycled Li metal in Fig. 4 shows that the SEI has a remarkable influence on the lithium plating behavior of the Li metal negative electrode. SEM images of Li metal surface after one cycle using the baseline electrolyte shows that Li plates and strips heterogeneously, leaving behind isolated Li islands (left panel in Fig. 4a). In comparison, the morphology of the Li metal

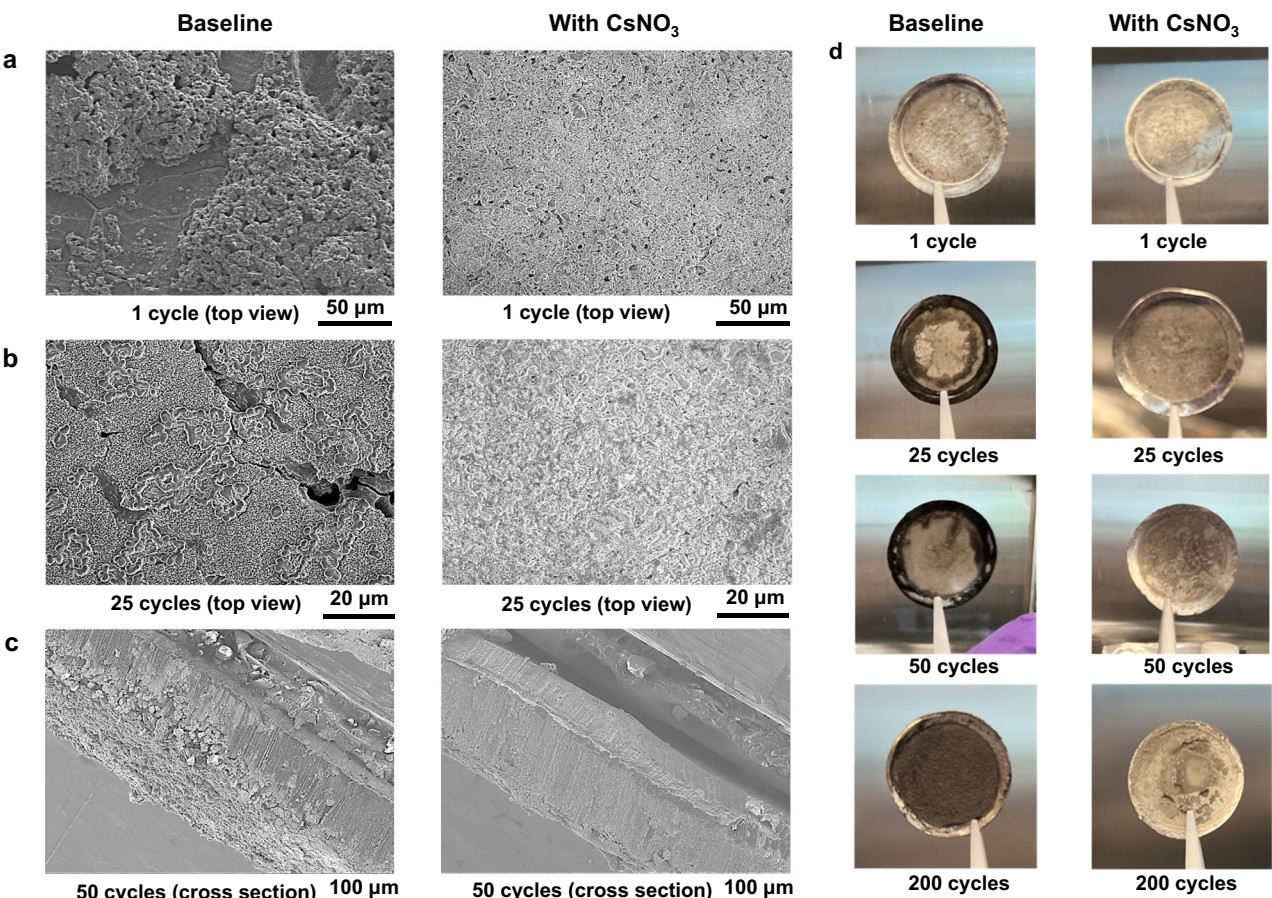

**Fig. 4 | Evidence of Li metal stabilization with CsNO₃ additive. a** Li metal morphology in baseline electrolyte (left) and in the electrolyte with 3 wt% CsNO₃ additive (right) after 1 cycle. **b** Li metal morphology in baseline electrolyte (left) and in the electrolyte with 3 wt% CsNO₃ additive (right) after 25 cycles. **c** Cross-sectional morphology of Li metal in the baseline electrolyte (left) and in the electrolyte with 3 wt% CsNO₃ additive (right) after 50 cycles. **d** Digital photographs of cycled Li metal with the baseline electrolyte (left) and in electrolyte with the CsNO₃ additive (right).

after one cycle using electrolyte with CsNO₃ additive is in general homogeneous and dense (right panel in Fig. 4a). Upon cycling, the inhomogeneity of Li plating becomes more obvious when baseline electrolyte is used. After 25 cycles, two kinds of distinct Li morphology can be observed on the Li metal surface using the baseline electrolyte (left panel in Fig. 4b): small grains of Li and large grains of Li islands. In comparison, deposited Li metal using electrolyte with CsNO₃ additive is dense and smooth without islands of large chunk of Li metal (right panel in Fig. 4b). Similar deposition morphology is maintained during prolonged cycling (Supplementary Fig. S18). The difference in Li morphology is more obvious from the cross-sectional SEM images of the cycled Li metal. Loosely bound grains of Li can be observed on the Li metal cycled in the baseline electrolyte after 50 cycles (left panel in Fig. 4c). On the contrary, compact and dense Li morphology with no loosely bound Li grains is observed on the Li metal cycled using the electrolyte with CsNO₃ additive (right panel in Fig. 4c). Digital photographs of the Li metal cycled for different cycle numbers in the baseline electrolyte show black deposits on the surface of the Li metal (left panels in Fig. 4d). In contrast, Li metal cycled in the electrolyte with CsNO₃ additive are free of such black deposits (right panels in Fig. 4d). The positive effect of Cs⁺ on lithium plating uniformity has previously been reported in the literature[32] and attributed to the electrostatic shielding effect of Cs⁺. Our SEM characterization results agree well with previous literature report but based on the SEI characterization discussed in previous section, we believe Cs⁺ is doing much more than the electrostatic shielding effect as commonly believed. It is likely that the superior interphase dominated by CsFSI (and free of LiF) contributes

to more uniform local current and hence more dense and homogeneous lithium deposition.

## NMC811 stabilization in presence of N-rich interphase

Successful protection of the NMC811 positive electrode is indicated by soft X-ray absorption spectroscopy (sXAS) and X-ray fluorescence (XRF) data, which characterize the transition metal reduction and dissolution (Fig. 5). sXAS is conducted to evaluate the oxidation state change of the transition metals on the NMC811 surface after electrochemical cycling. Ni L-edge shows that the oxidation state of surface Ni remains largely comparable with the pristine state after electrochemical cycling, and this is in general true for both the baseline electrolyte and the electrolyte with CsNO₃ additive (Fig. 5a). The reason may be that for high nickel NMC, the surface is already reduced in the pristine material as discussed in our previous work[19]. Co L-edge reveals that Co is considerably reduced on the NMC811 surface when the baseline electrolyte is used. However, when CsNO₃ additive is used in the electrolyte, the surface is protected, and Co reduction is suppressed (Fig. 5b). NMC811 surface protection may be best illustrated by the behavior of Mn on the surface. Figure 5c shows that Mn undergoes significant reduction upon cycling in the baseline electrolyte. In contrast, the Mn oxidation state mostly remains similar to the pristine state when CsNO₃-containing electrolyte is used. Therefore, the CsNO₃ additive effectively protects the NMC811 surface and suppresses transition metal reduction. Such stabilization is likely to further mitigate transition metal dissolution and subsequent deposition on the lithium metal side. Transition metal deposition on the cycled Li metal

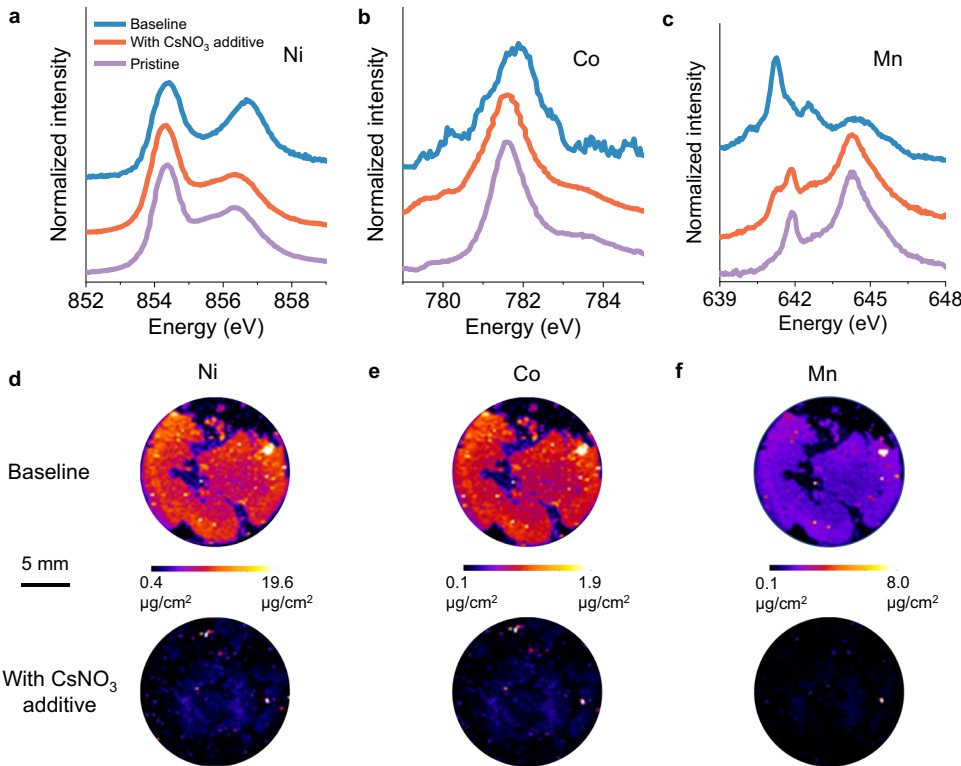

**Fig. 5 | Evidence of NMC811 stabilization with CsNO₃ additive. a** Ni, **b** Co, and **c** Mn L-edge soft X-ray absorption spectroscopy (sXAS) spectra of cycled NMC811 (100 cycles) with the baseline electrolyte and the electrolyte with CsNO₃ additive. X-ray fluorescence (XRF) imaging showing the distribution of **d** Ni, **e** Co, and **f** Mn on Li metal upon cycling in the baseline electrolyte (top) and the electrolyte with CsNO₃ additive (bottom) after 50 cycles.

is characterized by XRF imaging as shown in Fig. 5d–f. The distribution and the concentration of the deposited Ni species on the Li metal shows that Ni deposition on the Li metal is significantly suppressed by the addition of the CsNO₃ additive (Fig. 5d). Suppression of Co and Mn deposition on the Li metal is also evident as indicated by Fig. 5e, f. In fact, around 200 cycles are required for Li cells with the CsNO₃ additive to be able to observe around similar level of transition metal deposition in the baseline electrolyte after 50 cycles (Supplementary Fig. S19). The trend in transition metal deposition follows the cycle life of LMBs in these two electrolytes. Capacity fading may be contributed by material loss due to transition metal dissolution, as well as surface reconstruction, and impedance development resulting from transition metal dissolution[48]. These results suggest that the CsNO₃ additive successfully performed its designed functionality of protecting the NMC811 by suppressing transition metal reduction and dissolution.

## Discussions

In summary, we used CsNO₃ as an electrolyte additive for LMB and it can form good interphase on both the electrodes. The huge size of Cs⁺ leads to weak interaction between Cs⁺ cation and the solvents, allowing FSI anions to enter the solvation sheath and participating in the interphase formation. The resulting interphase is anion-derived as expected but surprisingly dominated by the inorganic species CsFSI. It is free of LiF, the component commonly believed to be necessary for a good interphase. Such interphase is highly stable and makes it possible to use DME as the single solvent for LMB electrolyte. The 1.5 M LiFSI in DME electrolyte with CsNO₃ additive enables stable cycling of LMB cell with very high NMC811 loading and low N/P ratio. Even when cycled at 2 C rate (corresponding to 8 mA/cm²), the cell shows good capacity retention within 200 cycles. Our results emphasize that the role of LiF in SEI may need to be reconsidered and the role of Cs⁺ in LMB electrolyte needs to be revisited.

## Methods

### Electrochemistry

NMC811 with two different loadings (around 9 mg/cm² and 21 mg/cm² active material loading) provided by CAMP facility at Argonne National Laboratory were utilized for battery testing. The composition of the low mass loading was 90% active material, 5% carbon black, and 5% polyvinylidene fluoride (PVDF). The composition of the high mass loading was 96% active material, 2% carbon black, and 2% PVDF. 50 μm thick Li metal with high loading NMC811 was utilized for practical loading battery testing. 250 μm thick Li metal with low mass loading NMC811 was utilized for battery testing under excess Li loading condition. Baseline electrolyte was 1.5 M LiFSI in dimethoxyethane (DME). Baseline electrolyte with additives were prepared by dissolving 3 wt% LiNO₃, 2 wt% NaNO₃, 3 wt% KNO₃, 3 wt% RbNO₃, and 3 wt% CsNO₃. Batteries were assembled with NMC811, Li metal, and 50 μL baseline electrolyte or electrolyte with additive. Celgard 2320 was used as the separator. The high N/P cells with baseline and with additives were cycled at C/2 rate (0.8 mA/cm²), preceded by two formation cycles at C/10 rate (0.16 mA/cm²). The low N/P cells were cycled at 1 C (4 mA/cm²) and 2 C (8 mA/cm²) rates. Cycling at these rates were preceded by two cycles at C/10 (0.4 mA/cm²) rate and 5 cycles at C/5 rate (0.8 mA/cm²). All the cycling was performed within 2.8–4.3 V. Li||Li symmetric cells were cycled with 250 μm thick Li metal at 1 mA/cm² cycling rate for a total capacity of 1 mAh/cm². All electrochemical cycling is performed at room temperature. Coulombic efficiency (CE) was measured by following the method proposed by Zhang and coworkers (Eq. (1))[47].

$$CE = \frac{nQc + Qs}{nQc + Qt} \qquad (1)$$

Here, Qc is the cycling capacity (1 mAh/cm²), Qt is the total deposited capacity (5 mAh/cm²), and Qs is the final stripping capacity.

Ionic conductivity of the electrolytes at room temperature was measured through electrochemical impedance spectroscopy using a cell of two platinum electrodes[49]. EIS measurements were performed from 1 Hz to 5 MHz range at open circuit potential of the cell. Resistivity (ρ) of the solution was measured through the following equation:

$$R = \rho\left(\frac{l}{A}\right) \tag{2}$$

Here, R is the solution resistance, ρ is the solution resistivity (the reciprocal of the value is ionic conductivity), l is the distance between the electrodes, and A is the area of the electrodes. The value of l and A were estimated through measuring the conductivity of a 1 M NaCl in deionized water solution.

## Materials characterization

X-ray diffraction (XRD) and pair distribution function (PDF) data was collected in beamline 28-ID-2 at National Synchrotron Radiation Lightsource II (NSLS II). The wavelength used for data collection was 0.1818 Å. After cycling, the interphase samples are collected by scrapping off from the Li metal and allowed to dry in the glovebox for about an hour. Interphase samples were then densely packed in a Kapton capillary tube (4 cm length) and sealed with epoxy glue on both ends (1 cm on each end) inside the glovebox. Data integration was performed in Dioptas[50], PDF data was extracted using PDFGetX3[51], and Rietveld refinement was performed in TOPAS software[52]. Every XRD data was collected 4 times, each with 5 minutes exposure time. Every PDF data was collected 5 times, each with 2 minutes exposure time. Cs L3-edge was collected from beamline 7-BM at NSLS II in fluorescence mode. The samples were sandwiched between Kapton tape for measurement. N K-edge and transition metal (TM) L-edges were acquired in the beamline 23-ID-2 in total electron yield mode. A pair of elliptical polarized undulators delivered X-ray of bright intensity for measurement with low energy X-ray. X-ray fluorescence (XRF) imaging was performed in beamline 5-ID at NSLS-II. Samples were taped between Kapton tape and Myler film, with Myler film facing the beam. Beam was focused with a pair of Kirkpatrick-Baez mirrors. Millimeter scale mapping of elements on electrode samples were performed with a step size of 200 μm. XRF data was fitted and quantified according to micromatter standards (Micromatter Technologies, Inc.) measurements with PyXRF software[53]. SEM imaging was performed in Hitachi SEM 4800 at 5 kV in secondary electron mode. All ex-situ samples were prepared inside the glovebox in Ar atmosphere. The cycled batteries were disassembled, electrodes harvested, washed with DME and dried before putting on the sample stage for TM L-edge measurement. Samples were then packed in an Ar atmosphere and delivered to the beamline. For interphase characterization through XRD, XRF, and XAS (Cs L3-edge and N K-edge), no washing was performed to preserve the interphase components.

## Data availability

The electrochemistry data is available at https://doi.org/10.6084/m9.figshare.24588552. All other data supporting the findings are available from the corresponding author upon request.

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

## Acknowledgements

The work at BNL is supported by the Assistant Secretary for Energy Efficiency and Renewable Energy (EERE), Vehicle Technology Office (VTO) of the US Department of Energy (DOE) through the Advanced Battery Materials Research (BMR) Program including the Battery500 Consortium under contract no. DE-SC0012704. This research used beamlines 5-ID, 7-BM, 23-ID-2, 28-ID-2 of the National Synchrotron Light Source II, a US DOE Office of Science user facility operated for the DOE Office of Science by Brookhaven National Laboratory under contract number DE-SC0012704. SEM measurements used the resources of the Center for Functional Nanomaterials, a US DOE Office of Science User Facility at BNL, under contract no. DE-SC0012704. We also acknowledge the US DOE CAMP (Cell Analysis, Modeling and Prototyping) Facility, Argonne National Laboratory for supplying the NMC811 electrodes. The CAMP Facility is fully supported by the DOE Vehicle Technologies Office.

## Author contributions

M.M.R. and E.H. conceived the idea. M.M.R. performed electrochemical characterization. M.M.R., Y.Y., H.Z., S.G., I.W., A.H., and L.M. performed soft and hard X-ray absorption spectroscopy and X-ray diffraction. S.T. performed SEM imaging. All authors contributed to the data analysis. M.M.R. and E.H. wrote the manuscript with input from all the coauthors.

## Competing interests

E.H. and M.M.R. report a US non-provisional patent application and a PCT application filed by Brookhaven National Laboratory on the basis of this work. Other authors do not have competing interests.
