## [Peer Review File · Nature Communications]

REVIEWER COMMENTS

Reviewer #1 (Remarks to the Author):

The work by Mohammad et al. reports using CsNO₃ as additive for full cell NMC | Li metal batteries. There are two interesting results in their work. One is the identification of an SEI that is free of LiF but mostly consists of CsFSI. This SEI seems to be a good one as the lithium stripping/plating efficiency is considerably high. Such finding may stimulate some new ideas in the battery community as currently it is well accepted that LiF is the key component for a good SEI. The second interesting result is the fast charging/discharging capabilities demonstrated for lithium metal batteries. It is appreciated that authors provide detailed parameters for their cell testing. The charging current for high cathode loading cells probably is one of the highest reported in the literature so far. While the performance is impressive, authors need to be sober-minded when making comparisons. This will be elaborated in the detailed comments. In general, this work is of good quality and can be considered for publication in Nature Communications after addressing the following comments.

1. Authors claim that there is no LiF in the lithium metal SEI but shows the presence of LiH. What is the basis of differentiating these two phases? They are isostructural and should have similar lattice parameters. In other words, how can the authors be sure that the peaks they label as LiH in Figure 2 are not from LiF? This is the most important question that authors need to address as this is the basis of some of the major claims made by the authors.
2. The presence of CsFSI in the SEI is interesting. It is noted that this component has two kinds of phases, the cis-CsFSI and the trans-CsFSI. Which one is more stable? Usually, if a material has cis- and trans- isomers, one phase would eventually transform to the other. Is this also the case for CsFSI?
3. What is the basis of using 1.5M rather than 1M for LiFSI concentration in the electrolyte? If it is based on the ionic conductivity, authors should provide the conductivity Vs. concentration plot to support their choice.
4. Authors should provide electrochemical window study like LSV or CV to show at what voltage does the electrolyte decompose and form the interphase.
5. There are some previous works also using the dual protection strategy for lithium batteries. A recent example is ACS Energy Letters 7 (9), 2866-2875. Authors should mention this work in their discussions.
6. The charging current used in this work is indeed very high. Figure 3f gives readers the impression that the performance in this work is far better than any literature report in terms of both charging current and cycle life, which may not be the case. There is a recent report by Ali Coskun's group (Nat. Commun. 2023, 14, 299) that shows stable cycling at a current density of 4.8 mA/cm². This data should be included in Figure 3f.

7. Authors quantified the amount of transition metals deposited on the lithium metal anode. Can authors have an estimate of how much transition metal is lost through the dissolution-deposition process? Would that be one of the causes for the NMC capacity loss?

Reviewer #2 (Remarks to the Author):

The manuscript submitted by Rahman et al. demonstrated the use of CsNO₃ as a dual-functional electrolyte additive that formed good interphases on both NMC cathode and Li metal anode to enable fast-charging capability and long cycle life of Li-metal-batteries. In addition, this work reported the presence of an inorganic-rich Cs containing interphase, suggesting a more enhanced role of Cs⁺ in the interphase formation. The paper also showed that the absence of LiF does not negatively influence the quality of the stable interphase formation, as commonly hypothesized in the research community.

The paper is scientifically robust and very well-written. Therefore, I recommend this paper to be published in Nature Communications after it has been improved with the following minor revisions:

1) On page 2, line 44 Define N/P for non-battery readers.

2) Are the electrodes in Figure 1c after formation cycles or after the 200 cycles? If after formation cycles, do the authors know how the Cs content on the anode changes with cycling (after 200 cycles)? What cycling rate was used for the 200 cycles? Was the XRD and PDF also done after 200 cycles?

3) Figure 2a, the stick-ball structures are too small to see easily when I printed the manuscript. Same for the text in a and b. Also, for Figure 2 a and b, is red the fit and blue the data for XRD and then red the data for PDF? The second (~1.9 Å) and third (~2.5 Å) peaks in the PDF of the baseline do not seem to fit well with any of the amorphous components suggested. What is the explanation for why the calculated CsFSI PDF doesn't match well with the data (red) for the with CsNO₃ sample? I'm not sure how helpful the SEI representation in Figure 2c is. What is the blue "sea" in which the SEI components reside represent?

4) Figure 3(c), it is hard to read "temperature fluctuation" text. Include the citation for the Aurbach method for CE calculations in the Figure 3 caption.

5) Mention briefly why is CE greater than 100 in Figures 3(d) and 3(e).

Reviewer #1 (Remarks to the Author):

The work by Mohammad et al. reports using CsNO₃ as additive for full cell NMC||Li metal batteries. There are two interesting results in their work. One is the identification of an SEI that is free of LiF but mostly consists of CsFSI. This SEI seems to be a good one as the lithium stripping/plating efficiency is considerably high. Such finding may stimulate some new ideas in the battery community as currently it is well accepted that LiF is the key component for a good SEI. The second interesting result is the fast charging discharging capabilities demonstrated for lithium metal batteries. It is appreciated that authors provide detailed parameters for their cell testing. The charging current for high cathode loading cells probably is one of the highest reported in the literature so far. While the performance is impressive, authors need to be sober-minded when making comparisons. This will be elaborated in the detailed comments. In general, this work is of good quality and can be considered for publication in Nature Communications after addressing the following comments.

Response: We thank the reviewer for the valuable comments.

1. Authors claim that there is no LiF in the lithium metal SEI but shows the presence of LiH. What is the basis of differentiating these two phases? They are isostructural and should have similar lattice parameters. In other words, how can the authors be sure that the peaks they label as LiH in Figure 2 are not from LiF? This is the most important question that authors need to address as this is the basis of some of the major claims made by the authors.

Response: Thank you for your comment. Although it is true that LiH and LiF are isostructural, they can be differentiated by different lattice parameters, which can be obtained through Rietveld refinement. The refined lattice parameter for the 'LiH' phase in the SEI XRD is 4.084 Å, matching values reported for LiH phase in the literature perfectly (*Physical Review B* **5**, 4704 (1972), *Journal of the American Chemical Society* **129**, 1594-1601 (2007), *Journal of crystal growth* **68**, 741-746 (1984)). Meanwhile, LiF has smaller lattice parameter, and the difference is significant enough to be differentiated through Rietveld refinement. Another difference between LiH and LiF in the SEI is the microstructure. LiH is usually in the form of large crystals (>200nm) while LiF is usually nano-crystals (around tens of nm) (*Nature* **560**, 345–349 (2018); *Energy & Environmental Science* **11**, 2600-2608 (2018)). More details about differentiating LiH and LiF in the SEI have been explained in our previous publications (*Nature Nanotechnology* **16**, 549-554 (2021), *Nature Nanotechnology* **18**, 243-249 (2023)). Based on the reviewer's comment, we have added the following discussion in the revised manuscript:

Page 8. It should be noted that LiH and LiF have the same crystal structure but different lattice parameters (*J. Am. Chem. Soc.* **129**, 1594-1601 (2007); *Phys. Rev. B* **5**, 4704 (1972); *J. Cryst. Growth* **68**, 741-746 (1984)). The difference is significant enough to be differentiated by Rietveld

refinement. When LiH and LiF are both present in the SEI, they usually have different microstructure properties with LiH being in the form of large crystals (>200 nm) while LiF being nanocrystals (~ 10 nm) (*Nature* **560**, 345-349 (2018); *Energy Environ. Sci.* **11**, 2600-2608 (2018). More details on differentiating LiH from LiF have been explained in our previous work (*Nat. Nanotechnol.* **16**, 549-554 (2021); *Nature Nanotechnology* **18**, 243-249 (2023)).

2. The presence of CsFSI in the SEI is interesting. It is noted that this component has two kinds of phases, the cis-CsFSI and the trans-CsFSI. Which one is more stable? Usually, if a material has cis- and trans- isomers, one phase would eventually transform to the other. Is this also the case for CsFSI?

Response: Thank you very much for your comment. It is true that isomorphs may have different thermodynamic stabilities. As a result, there could be an evolution of the distribution among cis- and trans- phases of CsFSI. To address the reviewer's question, we have performed additional experiments to characterize the SEI evolution during cycling with an emphasis on the CsFSI phase. The XRD data from different cycles reveal a dynamic evolution between the cis- and trans- CsFSI phases. After 100 cycles, the ratio between cis-CsFSI and trans-CsFSI is 35.2: 64.8. After 200 cycles, this ratio becomes 91.0: 9.0. It shows that over time the fraction of cis-CsFSI phase increases while that of the trans-CsFSI phase decreases, indicating a trans- to cis- phase transition and the cis- phase is likely the more thermodynamically stable one. This is consistent with previous findings reported in the literature (*Inorganic Chemistry* **52**, 568–576 (2013)) which suggests a spontaneous transition from trans-CsFSI to cis-CsFSI. The new results and the discussion are incorporated in the revised manuscript, which is copied below:

Page 9. Interestingly, there is a dynamic evolution of the two isomorphs of CsFSI upon cycling. Comparison between the SEI species after 100 cycles (Figure S3) and 200 cycles (bottom panel of Figure 2a) shows that upon cycling, the fraction of cis-CsFSI increases while that of trans-CsFSI decreases, indicating a trans-to-cis phase transition and the cis- isomorph being the more thermodynamically stable one. This is consistent with previous findings reported in the literature (*Inorg. Chem.* **52**, 568-576 (2013)).

Figure S3. Rietveld refinement of the SEI species of Li metal anode cycled in the CsNO₃ containing electrolyte after 100 cycles.

Table S3. Crystallographic data obtained by the Rietveld refinement of the XRD pattern of the SEI species after 100 cycles.

atom	site	x	y	z	occupancy	B value
Li space group: Im$\bar{3}$m a=b=c=3.511(1) Å phase fraction: 61.63%						
Li	2a	0	0	0	1	4.54
cis-CsFSI space group: P21/n a=7.753(1) Å, b=8.738(1) Å, c=10.085(1) Å phase fraction: 5.1%						
Cs1		0.287	0.155	0.395	1	4.03
N1		0.651	0.353	0.379	1	0.2
S1		0.746	0.336	0.514	1	4.83
S2		0.691	0.483	0.277	1	4.83
O1		0.754	0.472	0.591	1	2.49
O2		0.682	0.203	0.578	1	2.49
O3		0.869	0.479	0.233	1	2.49
O4		0.575	0.469	0.171	1	2.49
F1		0.935	0.302	0.476	1	0.2
F2		0.664	0.633	0.345	1	0.2
trans-CsFSI space group: P21/n a=7.898(1) Å, b=8.469(1) Å, c=10.679(1) Å phase fraction: 9.38%						
Cs1		0.150	0.254	0.413	1	3.53
N1		0.562	0.299	0.444	1	0.2
S1		0.636	0.399	0.342	1	5.14
S2		0.683	0.192	0.54	1	5.14
O1		0.503	0.496	0.279	1	2.39

O2		0.800	0.466	0.386	1	2.39
O3		0.808	0.106	0.488	1	2.39
O4		0.584	0.114	0.619	1	2.39
F1		0.670	0.277	0.241	1	0.2
F2		0.790	0.315	0.630	1	0.2

LiH space group: $Fm\bar{3}m$ $a=b=c=4.084(1)$ Å phase fraction: 10.67%

Li	4a	0	0	0	1	1.77
H	4b	0.5	0.5	0.5	1	2.0

LiOH space group: $P4/nmm:2$ $a=b=3.55(1)$ Å, $c=4.43(1)$ Å phase fraction: 13.21%

O	2c	0.25	0.25	0.179	1	4.937
Li	2a	0.75	0.25	0	1	0.407
H	2c	0.25	0.25	0.41	1	2

3. What is the basis of using 1.5M rather than 1M for LiFSI concentration in the electrolyte? If it is based on the ionic conductivity, authors should provide the conductivity Vs. concentration plot to support their choice.

Response: Thank you for raising this question. The 1.5 M LiFSI concentration was chosen based on optimized ionic conductivity and salt concentration. The aim was to get fairly high ionic conductivity without raising the concentration of LiFSI salt too much (so the cost can be relatively low). We determined the ionic conductivity at different LiFSI concentrations. The concentration vs ionic conductivity plot shows that the ionic conductivity of 1.5M LiFSI is not too far below 2M LiFSI. Furthermore, 1.5 M salt concentration does not put it in the high-concentration electrolyte range either. The ionic conductivity tends to go down above 2M LiFSI concentration. Hence, based on salt concentration and ionic conductivity, the 1.5 M concentration was chosen for further study. We have added the following additional data and discussion to clarify this point.

Page 13. Moreover, LiFSI in DME based electrolytes offer high ionic conductivity. For example, 1.5 M LiFSI salt concentration delivers reasonably high ionic conductivity without using too high salt concentration (Figure S12).

Figure S12. Concentration vs ionic conductivity plot of various LiFSI in DME electrolytes.

4. Authors should provide electrochemical window study like LSV or CV to show at what voltage does the electrolyte decompose and form the interphase.

Response: Thank you very much for the suggestion. We have performed CV in Li||NMC cells (0 to 5V range) to determine the stability window. We performed the analysis in Li||NMC because the determination of decomposition potential in Li||Al or Li||Cu cell tends to overestimate the real voltage stability window. The presence of NMC can catalyze electrolyte decomposition, leading to a narrower (but practically more relevant) electrochemical window (*Angewandte Chemie International Edition*, e202311051 (2023)). The first cycle CV shows that the high voltage decomposition peak is around 4.88 V in both 1.5 M LiFSI in DME electrolyte and the electrolyte containing the CsNO₃ additive. Meanwhile, the low voltage decomposition potential in the Li||NMC811 can be found at around 0 V. The data and discussion are added in the manuscript and copied below:

Page 12. Both the baseline electrolyte and the electrolyte with the CsNO₃ additive show comparable oxidative/reductive decomposition potential (Figure S6). However, the interphase compositions resulting from the decomposition of these two electrolytes are significantly different as discussed in Figure 2.

Figure S6. Cyclic voltammetry curves of Li||NMC811 cells cycled in 1.5 M LiFSI in DME (baseline) electrolyte and electrolyte with CsNO₃ additive.

5. There are some previous works also using the dual protection strategy for lithium batteries. A recent example is ACS Energy Letters 7 (9), 2866-2875. Authors should mention this work in their discussions.

Response. Thank you for drawing our attention to this work. We have added further discussion on the dual protection strategy acknowledging previous works adopting this strategy. While many works achieve the dual protection through combination of additives or a combination of fluorinated solvents and additives, our work aims to achieve this through utilizing a single additive that can act upon both electrodes.

Page 3. This emphasizes the dual protection strategy. Several studies have explored the strategy with various electrolyte additives (*Nat. Commun.* **13**, 1297 (2022), *ACS Energy Lett.* **7**, 2866-2875 (2022)) We will further explore the strategy through rational design of single additive that can simultaneously act upon both electrodes to promote the formation of stable interphases.

6. The charging current used in this work is indeed very high. Figure 3f gives readers the impression that the performance in this work is far better than any literature report in terms of both charging current and cycle life, which may not be the case. There is a recent report by Ali Coskun's group (*Nat. Commun.* 2023, 14, 299) that shows stable cycling at a current density of 4.8 mA/cm². This data should be included in Figure 3f.

Response: Thank you very much for drawing our attention to this work. We have included the data from Ali Coskun's work in Figure 3f. The new figure is copied below:

It should be noted that two additional data points from the recommended work were added in the plot. Data point 63a utilized a cathode loading of 20 mg/cm² whereas datapoint 63b uses 8 mg/cm².

7. Authors quantified the amount of transition metals deposited on the lithium metal anode. Can authors have an estimate of how much transition metal is lost through the dissolution-deposition process? Would that be one of the causes for the NMC capacity loss?

Response: Thank you for raising this question. Transition metal dissolution is attributed as one of the key reasons of cathode capacity loss. However, it is difficult to accurately ascertain the extent of transition metal dissolution from the cathode because of the complex cathode microstructure. Estimating transition metal dissolution only from the separator may not accurately reflect the amount of transition metal dissolution from the cathode. Nonetheless, a relative estimation of transition metal dissolution can still be provided through mapping the distribution of the transition metals on the anode side through X-ray fluorescence (XRF) mapping. Transition metals from the cathode find their way onto the anode through dissolution and subsequent migration through the electrolyte. XRF mapping allows to compare the relative extent of transition metal dissolution on cells cycled in different electrolyte by comparing the extent of transition metal deposition on the anode side. Our analysis shows that the extent of transition metal deposition follows the expected life of batteries cycled in the baseline electrolyte and the electrolyte containing the CsNO₃ additive. Hence, transition metal dissolution can be attributed as one of the contributing reasons of battery capacity fading. Although transition metal dissolution indicates material loss which contributes to capacity fading, the consequence of transition metal dissolution can be multifaceted. Previous reports suggest that the greater contributing factor to cathode capacity loss comes from the alteration of cathode surface structure and rise in cathode impedance development due to transition

metal dissolution (*Energy & Environmental Science* **11**, 243-257 (2018)). We have added the following discussion based on this comment:

Page 18. The trend in transition metal deposition follows the cycle life of LMBs in these two electrolytes. Capacity fading may be contributed by both material loss due to transition metal dissolution, as well as cathode surface reconstruction, and impedance development resulting from transition metal dissolution (*Energy Environ. Sci.* **11**, 243-257 (2018)).

Reviewer #2 (Remarks to the Author):

The manuscript submitted by Rahman et al. demonstrated the use of CsNO₃ as a dual-functional electrolyte additive that formed good interphases on both NMC cathode and Li metal anode to enable fast-charging capability and long cycle life of Li-metal-batteries. In addition, this work reported the presence of an inorganic-rich Cs containing interphase, suggesting a more enhanced role of Cs⁺ in the interphase formation. The paper also showed that the absence of LiF does not negatively influence the quality of the stable interphase formation, as commonly hypothesized in the research community.

The paper is scientifically robust and very well-written. Therefore, I recommend this paper to be published in Nature Communications after it has been improved with the following minor revisions:

1) On page 2, line 44 Define N/P for non-battery readers.

Response: thank you for the comment. We have defined the N/P ratio as suggested by the reviewer.

Page 2. These novel electrolytes have enabled LMB with long cycle even under harsh conditions such as high cathode loading and low N/P ratio (defined as capacity ratio between cathode and anode).

2) Are the electrodes in Figure 1c after formation cycles or after the 200 cycles? If after formation cycles, do the authors know how the Cs content on the anode changes with cycling (after 200 cycles)? What cycling rate was used for the 200 cycles? Was the XRD and PDF also done after 200 cycles?

Response: Thank you for raising the question. Cs mapping in Figure 1c was performed on the cathode and the Li metal after 200 cycles at C/2 rate. The PDF and XRD presented in Figure 2 are also done after 200 cycles. We have clarified the cycle numbers at relevant places in the manuscript. To show the evolution of Cs content on the interphase, we performed additional

experiments and mapped the distribution of Cs on Li metal anode after 50 cycles. The new data shows that Cs accumulation keeps happening from 50 cycles (around $60 \mu\text{g}/\text{cm}^2$) to 200 cycles (around $90 \mu\text{g}/\text{cm}^2$). However, most of the Cs accumulation on the anode interphase takes place within the first 50 cycles. The additional data is added in the manuscript and copied below:

Figure S1. X-ray fluorescence map of Cs distribution on Li metal anode after 50 cycles. The cycling was performed at C/2 rate.

Page 5. The accumulation of Cs is gradual, but majority of the Cs accumulates on the anode interphase by the first 50 cycles (Figure S1).

3) Figure 2a, the stick-ball structures are too small to see easily when I printed the manuscript. Same for the text in a and b. Also, for Figure 2 a and b, is red the fit and blue the data for XRD and then red the data for PDF? The second ($\sim 1.9 \text{ \AA}$) and third ($\sim 2.5 \text{ \AA}$) peaks in the PDF of the baseline do not see to fit well with any of the amorphous components suggested. What is the explanation for why the calculated CsFSI PDF doesn't match well with the data (red) for the with CsNO₃ sample? I'm not sure how helpful the SEI representation in Figure 2c is. What is the blue "sea" in which the SEI components reside represent?

Response: We apologize for the inconvenience. We have enlarged the stick-ball structures in the modified figure 2 for better clarity. In Figure 2a, the red is the fit and the blue is the XRD pattern for individual phases calculated from Rietveld refinement. In Figure 2b, the red is the measured PDF data and the blue is the PDF pattern calculated from the possible individual components of the SEI. The molecular/crystal structure of these components are optimized through DFT

calculations, which had been done in our previously published work (*Nat. Nanotechnol.* **16**, 549-554 (2021)). To avoid confusion and to explain the figures clearly, we have revised the figures and modified the captions accordingly.

We apologize for the confusion caused on the PDF peaks in Figure 2b. This kind of impression probably was caused by the scale of the component PDF which is relatively small. In the revised manuscript, we increased the scale of the component PDF and we believe now the match between measured PDF (dots) and the component PDF (red lines) may be much more obvious. During the revision, we also re-calculated the structure of these possible SEI components using DFT with a higher level of accuracies. However, it should be mentioned that the real species in the SEI may still differ slightly from the calculated ones because factors like defects and surface adsorbents can all affect the eventual structure. Unfortunately, those factors cannot be fully considered in the DFT calculation. Figure 2 has been modified for clarity and copied below:

Figure 2. Crystalline and amorphous components in lithium metal anode interphase. (a) Crystalline components of the anode interphase revealed by synchrotron XRD and Rietveld refinement. The red is the fit and the blue is the XRD pattern for individual phases calculated from Rietveld refinement. (b) Amorphous components of the anode interphase revealed by PDF studies. The scattered plot is the measured PDF data, and the red is the PDF pattern calculated from the possible individual components of the SEI. SEI species were characterized after 50 cycles in the baseline electrolyte and after 200 cycles in the electrolyte containing the CsNO₃ additive. The cycling was performed at C/2 rate. The color code for the atoms is the same in all subfigures: red:

oxygen; blue: nitrogen; cyan: fluorine; brown: carbon; white: hydrogen; gold: lithium; dark yellow: cesium, light yellow: sulfur.

Thank you for drawing our attention to the calculated PDF of CsFSI. The PDF peak at around 3.3 Å can only be accounted for by the CsFSI phase. The low r peak at around 1.5 Å cannot be fully accounted for by only considering CsFSI phase. This is because there are also FSI-derived species in the SEI that contribute to this peak.

We apologize for the inconvenience with the schematic illustration presentation in Figure 2c. The blue sea represents the amorphous matrix in the illustration. At the reviewer's suggestion, we moved Figure 2c to the supporting information.

4) Figure 3(c), it is hard to read "temperature fluctuation" text. Include the citation for the Aurbach method for CE calculations in the Figure 3 caption.

Response: We apologize for the inconvenience. We have enlarged the font size of the “temperature fluctuation” text and added the citation for the Aurbach method in the figure caption. The modified figure is copied below:

Figure 3. Electrochemical cycling performance of Li metal batteries in baseline electrolyte and with CsNO₃ additive. (a) Coulombic efficiency measurement following the Aurbach method in a Li||Cu cell (*Adv. Energy Mater.* **8**, 1702097 (2018)). (b) Cycling stability of Li metal cell with NMC811 cathode and Li metal anode with baseline electrolyte and electrolyte containing 3 wt% CsNO₃ additive. NMC811 loading is 9 mg/cm² and Li metal anode used is 250 μm thick. Cycling

is performed at C/2 rate. (c) Cycling stability of Li metal cell using NMC811 cathode with 21 mg/cm² loading and 50 μm thick Li metal anode in the electrolyte containing 3 wt% CsNO₃ additive. The cells go through two formation cycles at a rate of C/10 and five cycles at C/5 rate before being cycled at 1C rate and 2C for later cycles. (d) Coulombic efficiency of the cell in Figure 3b. (e) Coulombic efficiency of the cells in Figure 3c. (f) Comparison of charge current density and cycle life of LMB reported in the literature with this work (*Nat. Energy* **7**, 94-106 (2022); *Joule* **3**, 1662-1676 (2019); *Nat. Energy* **5**, 526-533 (2020); *Nat. Energy* **8**, 340-350 (2023); *Energy Environ. Sci.* **15**, 1907-1919 (2022); *Nat. Energy* **6**, 495-505 (2021); *Angew. Chem. Int. Ed.* **61**, e202115884 (2022); *Nat. Nanotechnol.* **13**, 715-722 (2018); *Proc. Natl. Acad. Sci.* **115**, 1156-1161 (2018); *Angew. Chem. Int. Ed.* **59**, 14935-14941 (2020); *Energy Environ. Sci.* **15**, 2435-2444 (2022); *Adv. Funct. Mater.* **30**, 2003800 (2020); *Energy Environ. Sci.* **15**, 4349-4361 (2022); *Nat. Energy* **7**, 548-559 (2022); *Energy Environ. Sci.* **13**, 212-220 (2020); *Energy Storage Mater.* **34**, 76-84 (2021); *Adv. Mater.* **32**, 2001740 (2020); *Nat. Commun.* **14**, 299 (2023)). Data points for ref. 63 (*Nat. Commun.* **14**, 299 (2023)) utilize two different cathode loadings. 63a: 20 mg/cm² and 63b: 8 mg/cm².

5) Mention briefly why is CE greater than 100 in Figures 3(d) and 3(e).

Response: Thank you for drawing our attention to this issue. Indeed, the CE in figure 3d and 3e in certain cycles randomly goes above 100%. Even though CE is routinely measured and conventionally regarded as the indicator of efficiency, it should be noted that the real meaning of CE is the number of electrons flow during discharged divided by the electrons flow during charge. Due to the presence of side reactions, there is no direct correlation between the number of electrons used and the number of lithium ions transported. As a result, CE may not be a good indicator of efficiency of lithium metal batteries and the randomness of side reactions during charge-discharge may give rise to CE number over 100%. For example, more electrons could be consumed during discharge than during charge because of more side reactions and this would lead to a CE greater than 100%. A previous paper by Xiao et al has discussed this phenomenon in details (*Nature energy*, **5**, 561-568 (2020)). We made the following revisions with regard to this point:

Page 14. In some cycles (both in Figure 3d and 3e), the CE may even go up over 100%. Even though CE is routinely measured and conventionally regarded as the indicator of efficiency, it should be noted that the real meaning of CE is the number of electrons flow during discharged divided by the electrons flow during charge. Due to the presence of side reactions, there is no direct correlation between the number of electrons flow and the number of lithium ions transported. As a result, CE may not be a good indicator of efficiency for LMB and the randomness of side reactions during charge-discharge may give rise to CE number over 100%. For example, more electrons could be consumed during discharge than during charge because of more side reactions and this would lead to a CE greater than 100%.

REVIEWERS' COMMENTS

Reviewer #1 (Remarks to the Author):

I have checked the authors' responses and believed that all concerns have been well addressed. Now, I recommend the acceptance of this work by Nature Communications.